# AdwU-Net: Adaptive Depth and Width U-Net for Medical Image Segmentation by Differentiable Neural Architecture Search

**Ziyan Huang**[1]                                          ZIYANHUANG@SJTU.EDU.CN
[1] *Institute of Medical Robotics, Shanghai Jiao Tong University*

**Zehua Wang**[2]                                          WANGZH@CMDE.ORG.CN
[2] *Center for Medical Device Evaluation, National Medical Products Administration*

**Zhikai Yang**[1]                                          XIAOERLAIGEID@SJTU.EDU.CN
**Lixu Gu**[1]                                          GULIXU@SJTU.EDU.CN

**Editors:** Under Review for MIDL 2022

## Abstract

The U-Net and its variants are proved as the most successful architectures in the medical image segmentation domain. However, the optimal configuration of the hyperparameters in U-Net structure such as depth and width remain challenging to adjust manually due to the diversity of medical image segmentation tasks. In this paper, we propose AdwU-Net, which is an efficient neural architecture search framework to search the optimal task-specific depth and width in the U-Net backbone. Specifically, an adaptive depth and width block is designed and applied hierarchically in U-Net. In each block, the optimal number of convolutional layers and channels in each layer are directly learned from data. To reduce the computational costs and alleviate the memory pressure, we conduct an efficient architecture search and reuse the network weights of different depth and width options in a differentiable manner. Extensive experiments on the Medical Segmentation Decathlon (MSD) dataset show that our method outperforms not only the manually scaled U-Net but also other state-of-the-art architectures. Our code is publicly available at https://github.com/Ziyan-Huang/AdwU-Net.

**Keywords:** Medical image segmentation, neural architecture search, model scaling.

## 1. Introduction

The vast majority of successful algorithms for medical image segmentation are based on the U-Net architecture (Ronneberger et al., 2015). However, the diversity of medical image segmentation tasks could be extremely high, since dataset properties including image size, image spacing, voxel intensity range, intensity interpretation and anatomical regions of interest can vary considerably (Antonelli et al., 2021). Therefore, the optimal architectural hyperparameters including depth and width can vary from one application to another. To achieve better segmentation performance, it is highly desirable for task-specific hyperparameters design for U-Net.

Recently, a few studies have noticed the different optimal depth and width of U-Net for different tasks. U-Net++ (Zhou et al., 2020) ensembles U-Nets of varying downsampling

times to alleviate the unknown depth. nnU-Net (Isensee et al., 2021) configures the number of downsampling operations along each axis depending on the patch size and voxel spacing. The width (a.k.a. the number of channels or the number of filters) in U-Net is another important hyperparameter that has been chosen mostly based on heuristics. Most works only consider the width of the first stage manually (Ronneberger et al., 2015; Zhou et al., 2020; Isensee et al., 2021) or automatically (Calisto and Lai-Yuen, 2021). Then, the "*half size, double channel*" rule is adopted by the rest of the U-Net layers. However, the optimal depth and width in each resolution stage of U-Net are rarely considered. Moreover, (Tan and Le, 2019) suggest that there is a high correlation between optimal depth and width, aforementioned works don't consider depth and width of U-Net simultaneously.

Inspired by Automated Machine Learning (AutoML), there has been great interest in neural architecture search (NAS). Previous NAS methods are based on reinforcement learning (RL) (Zoph et al., 2018) or evolutionary algorithms (EA) (Real et al., 2019), leading to the excessive consumption of computing resources. Differentiable neural architecture search (DNAS) (Liu et al., 2019) reduce the search cost to several GPU days by coupling architecture sampling and training into a super-network. Recently, some works apply NAS to medical image segmentation. (Weng et al., 2019) and (Kim et al., 2019) search the architecture of cells and stacked them repeatedly in a U-Like backbone network. (Zhu et al., 2019) let the network choose automatically between 2D, 3D, and pseudo-3D (P3D) convolutions for each layer. (Yu et al., 2020) and (He et al., 2021) search the topology of the network. (Ji et al., 2020) and (Yan et al., 2020) explore the feature aggregation strategies between different resolution scales. However, these methods still rely on the human choice of network depth and width, which can lead to sub-optimal solutions.

In this paper, we propose the adaptive depth and width U-Net (AdwU-Net), which is a differentiable neural architecture search framework that can adaptively adjust the depth and width of U-Net in different medical image segmentation tasks. Our main idea is to construct the adaptive depth and width block (AdwBlock) for each resolution stage in the U-Net backbone. For depth search, we design a sink-connecting search space where the outputs of all layers in a stage are connected to a sink point. For width search, we represent multiple channel convolutions with only one convolution with multiple masks. We evaluate the effectiveness of our proposed method on the Medical Segmentation Decathlon (MSD) dataset. Compared to U-Net models with manually designed depth and width, models with our searched depth and width are more efficient and effective. Meanwhile, we configure nnU-Net with our searched depth and width and verify the superiority of our searched models over other state-of-the-art architectures. Experimental results show the power of choosing task-specific optimal depth and width in the U-Net.

Our contributions can be summarized as the following three folds:

- We propose a new segmentation framework, Adaptive depth and width U-Net (AdwU-Net), which can efficiently complete optimal depth and width search for U-Net.

- We design a memory-efficient search space for depth and width search by feature map reusing, which enables us to explore more depth and width options.

- We apply our searched task-specific architectures in the nnU-Net framework and achieve state-of-the-art performance on the MSD dataset.

## 2. Adaptive Depth and Width U-Net

Our segmentation network follows the encoder-decoder structure. The depth and width in each stage of both encoder and decoder are learned in a differentiable way. The whole network structure is illustrated in Figure 1. We use U-Net as our backbone network and adopt the adaptive depth and width block in each resolution stage.

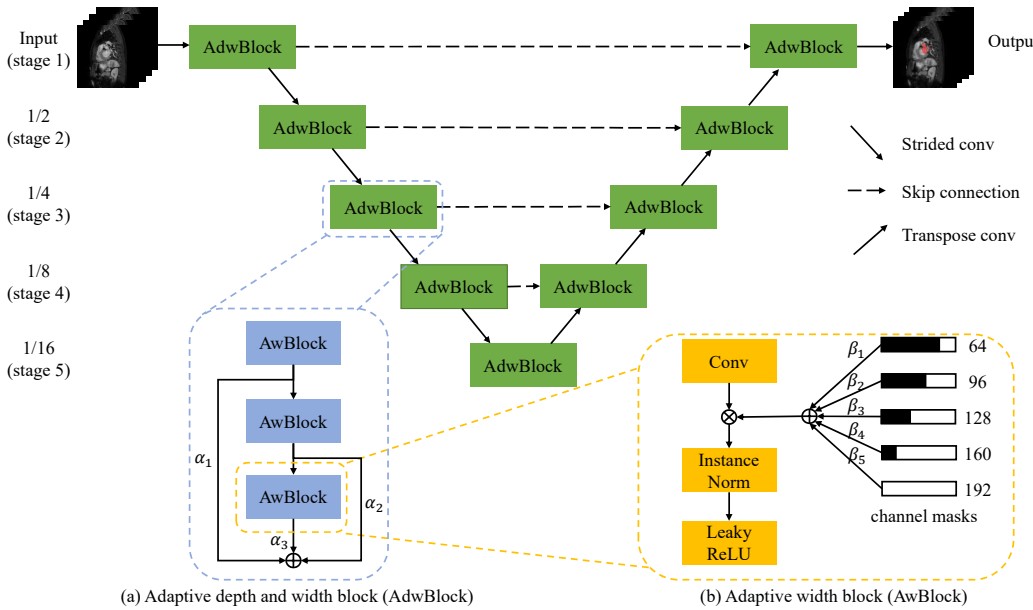

Figure 1: Overview of the proposed adaptive depth and width U-Net for medical image segmentation. (a) Adaptive depth and width block (AdwBlock) searches the optimal number of adaptive width blocks for each resolution stage. (b) Adaptive width block (AwBlock) searches the optimal width for each convolutional layer by masking.

### 2.1. U-Net Backbone

U-Net is a symmetric encoder-decoder architecture where both encoder and decoder can be divided into several stages of different resolutions. The encoder downsamples feature maps and extracts features from high resolution to low resolution, then the decoder upsamples feature maps and extracts features from low resolution to high resolution. Skip connections are used to fuse the feature maps in the same resolution of the output of the encoder and the input of the decoder.

### 2.2. Adaptive Depth and Width Block

We design the adaptive depth and width block (AdwBlock) for each resolution stage. As shown in Figure 1(a), each AdwBlock consists of three adaptive width blocks (AwBlock).

The AdwBlock is designed for the search of the optimal number of AwBlocks, which is also the optimal depth in each block. The AwBlock is designed for the search of optimal channel number of convolutional layers. For the depth level, each block can choose the number of convolutional layers between 1 to 3. For the width level, each convolutional layer can have 5 filter number options. Therefore, each AdwBlock has $5 + 5^2 + 5^3 = 155$ different candidate architectures. Consider the U-Net backbone with 11 blocks, then the AdwU-Net has $155^{11} \approx 10^{24}$ candidate architectures which are impossible to explore manually.

### 2.2.1. DEPTH SEARCH

For depth search, we search the optimal convolution number in each resolution stage. In the search procedure, the output of each resolution stage is the weighted sum of the outputs of different depth options. The naive implementation is to construct three independent parallel paths of different convolution numbers. However, this implementation will store all the feature maps of different paths in GPU memory separately, leading to a quadratic memory consumption w.r.t. the maximum candidate depth $L$ in the block. To avoid redundancy, we only keep the deepest path and add weighted skip connections from the output of preceding layers to the sink point at the end of each block as shown in Figure 1(a). In this implementation, the deeper path reuses the convolution weights of the shallower path, therefore we only need to compute the feature map of the deepest path.

Let $\alpha_l^s$ be the architecture parameter of $l_{th}$ depth option in stage $s$. We employ Gumbel Softmax (Jang et al., 2016) function as the continuous relaxation, it can be expressed as:

$$g_l^s = \frac{\exp[(\alpha_l^s + \epsilon_l^s)/\tau]}{\sum_l \exp[(\alpha_l^s + \epsilon_l^s)/\tau]} \tag{1}$$

where $\epsilon_l^s \in Gumbel(0,1)$ is a random noise following the Gumbel distribution and $\tau$ is a temperature parameter which decreases during the search procedure, forcing $g_l^s$ to approach a one-hot distribution.

Given the input $x^s$, the output of stage s is the weighted sum of the output of each convolutional layer:

$$y^s = \sum_l g_l^s o_l^s(x^s) \tag{2}$$

where $o_l^s$ is the output of $l_{th}$ layer in stage s.

The computational budget of the whole block during the search is roughly the same as computing the feature maps of the deepest path only once, which enables us to efficiently search the depth of each block in a differentiable manner.

### 2.2.2. WIDTH SEARCH

For width search, we search the optimal channel number of each convolutional layer. Compared to depth search, there are two problems in width search which are harder to handle by the DNAS method. First, the output dimensions of blocks with different channel numbers mismatch with each other, so we can't do a weighted summation of them directly. Second, the number of candidate width options is very large. Simply instantiating each convolution with a different channel option will cause high computational costs and memory issues.

Drawing inspiration from FBNetV2 (Wan et al., 2020), we represent convolutions with varying channel numbers by convolutions with equal channel numbers multiplied by different channel masks. Then, we share the weights of different convolutions to reduce computational costs and GPU memory consumption. Furthermore, the width search is simplified to only one convolution with the maximum candidate channel number and multiplied by the weighted summation of different channel masks as shown in Figure 1(b).

We use the output of preceding layer $o_{l-1}^s$ as the input of $l_{th}$ layer in stage $s$. We only run the convolution operation once, then multiply by the weighted summation of masks. Instance normalization (IN) and Leaky ReLU are applied after the convolutional layer.

$$z_l^s = conv(o_{l-1}^s) \odot \sum_{i=1}^{k} g_{l,i}^s M_{l,i}^s \tag{3}$$

$$o_l^s = \sigma(IN(z_l^s)) \tag{4}$$

where $M_{l,i}^s \in \mathbb{R}^k$ is a column vector which has ones in the leading i entries and zeros at the end, $g_{l,i}^s$ is the Gumbel weight parameter of the $i_{th}$ mask in the $l_{th}$ layer and $\sigma$ denotes Leaky ReLU.

In our experiment setting, there are 5 candidate channel numbers in each convolutional layer. The detailed configurations are listed in Table 1. The channel number at the first stage ranges from 16 to 48 with step 8. In the following stages, when the resolution is reduced to half, all of the 5 candidate channel numbers double. To avoid too many parameters and reduce the computational costs, the channel configurations in stages greater than 4 still follow the configuration of stage 4.

Table 1: Width search space for each resolution stage. Tuples of three values represent the lowest value, highest value, and steps between options (low, high, steps).

| Stage | Scale | Channel configuration |
|:-----:|:-----:|:---------------------:|
| 1 | 1 | (16, 48, 8) |
| 2 | 1/2 | (32, 96, 16) |
| 3 | 1/4 | (64, 192, 32) |
| 4 | 1/8 | (128, 384, 64) |
| 5 | 1/16 | (128, 384, 64) |

### 2.3. Optimization

Our proposed AdwU-Net comprises architecture parameters $\alpha$, $\beta$ and the network weight $w$. In the search procedure, the relaxation method introduced in the previous section makes it possible to jointly learn the architecture parameters and network weights. We adopt the mixed-level optimization strategy (He et al., 2020) as it can embed more information to update architecture parameters. We divide the training set into two disjoint parts: $trainA$ and $trainB$ to avoid the overfitting of architecture parameters to training data. In each iteration, we fix the network weight $w$ first and update architecture parameters $\alpha$ and $\beta$

using $trainA$ and $trainB$ in succession, then we fix the architecture parameters $\alpha$ and $\beta$ and update the network weight $w$ using $trainA$. We use the sum of dice loss and cross-entropy loss as our loss function.

$$L = L_{Dice} + L_{CE} \tag{5}$$

After searching, we obtain the optimal depth and width for each stage by *argmax* operation, that is we retain the depth and width options with the maximum architecture parameters. It is worth noting that the final architecture does not employ masking or require skip connection from preceding layers.

## 3. Experiments and Results

We conduct experiments on the 3D Medical Segmentation Decathlon[1] (MSD) dataset (Antonelli et al., 2021), which is a comprehensive benchmark to evaluate the generalizabily of medical image segmentation algorithms. It contains 10 segmentation tasks from different organs, modalities, imaging sources. These tasks are diverse and covering a large span of challenges in clinical. We report all results on the test set in terms of the Dice Coefficient (DSC) and higher score indicates a better result.

### 3.1. Implementation Details

We implement our method in the nnU-Net framework and adopt the same preprocessing, data augmentation, and postprocessing procedure as in nnU-Net. In all our experimental settings, the batch size is fixed to 2 and each epoch contains 250 iterations. In the search stage, we split the training set into two equal disjoint parts $trainA$ and $trainB$. We use SGD optimizer with learning rate 1e-2, Nesterov momentum 0.99, weight decay 1e-3 for network weights $w$. In the first 100 epochs, we train $w$ without updating architecture. The architecture parameters $\alpha$ and $\beta$ are initialized to 0. We set the initial temperature $\tau$ to 5.0 and exponentially anneal it by 0.97 for every epoch. In the following 150 epochs, we alternatively optimize $w$ with the SGD optimizer mentioned above and $\alpha$, $\beta$ with Adam optimizer (learning rate of 4e-4, weight decay 0). The search procedure takes 2 days on 1 NVIDIA V100 GPU with 32GB memory. After searching, we retrain the network with the searched depth and width of 1000 epochs for validation. We use SGD optimizer with Nesterov momentum 0.99, weight decay 1e-3. The learning rate starts at 0.01 and is decayed following the poly learning rate policy: $(1 - epoch/1000)^{0.9}$.

### 3.2. Segmentation Results on MSD

We apply the 5-fold cross-validation for each task and ensemble the results of models with a majority voting for the final prediction. In Table 2, we compare our AdwU-Net with state-of-the-art methods on all 10 tasks of the MSD dataset. (Kim et al., 2019; Yu et al., 2020; He et al., 2021) are NAS-based methods and design architectures automatically. These methods mainly focus on searching the different operations in each block and connection strategies between blocks. However, these methods don't consider the optimal depth and width design of networks. Our proposed AdwU-Net achieves best performance in 6 of 10

---

1. http://medicaldecathlon.com/

Table 2: Comparison with state-of-the-art methods on the MSD challenge test set by DSC. The results are obtained from the MSD test leaderboard.

| Task | Brain Tumor | | | | Heart | Liver | | |
|---|---|---|---|---|---|---|---|---|
| Class | 1 | 2 | 3 | Avg. | 1 | 1 | 2 | Avg. |
| nnU-Net(Isensee et al., 2021) | 0.6804 | 0.4681 | 0.6846 | 0.6110 | 0.9330 | **0.9575** | 0.7597 | 0.8586 |
| Kim et al(Kim et al., 2019) | 0.6740 | 0.4575 | 0.6826 | 0.6047 | 0.9311 | 0.9425 | 0.7296 | 0.8361 |
| C2FNAS(Yu et al., 2020) | 0.6762 | 0.4860 | 0.6972 | 0.6198 | 0.9249 | 0.9498 | 0.7289 | 0.8394 |
| DiNTS(He et al., 2021) | **0.6928** | **0.4865** | **0.6975** | **0.6256** | 0.9299 | 0.9535 | 0.7462 | 0.8499 |
| AdwU-Net (ours) | 0.6814 | 0.4717 | 0.6893 | 0.6141 | **0.9334** | 0.9519 | **0.7792** | **0.8656** |

| Task | Hippocampus | | | Prostate | | | Lung | Spleen |
|---|---|---|---|---|---|---|---|---|
| class | 1 | 2 | Avg. | 1 | 2 | Avg. | 1 | 1 |
| nnU-Net(Isensee et al., 2021) | 0.8164 | 0.8869 | 0.8946 | **0.7659** | **0.8962** | **0.8311** | 0.7397 | **0.9743** |
| Kim et al(Kim et al., 2019) | 0.9011 | 0.8872 | 0.8942 | 0.7264 | 0.8902 | 0.8083 | 0.6310 | 0.9192 |
| C2FNAS(Yu et al., 2020) | 0.8937 | 0.8796 | 0.8867 | 0.7488 | 0.8875 | 0.8182 | 0.7044 | 0.9628 |
| DiNTS(He et al., 2021) | 0.8991 | 0.8841 | 0.8916 | 0.7537 | 0.8925 | 0.8231 | 0.7475 | 0.9698 |
| AdwU-Net (ours) | **0.9044** | **0.8894** | **0.8969** | 0.7438 | 0.8942 | 0.8190 | **0.7602** | 0.9742 |

| Task | Pancreas | | | Hepatic Vessel | | | Colon | Overall |
|---|---|---|---|---|---|---|---|---|
| class | 1 | 2 | Avg. | 1 | 2 | Avg. | 1 | Avg. |
| nnU-Net(Isensee et al., 2021) | 0.8164 | 0.5278 | 0.6721 | **0.6646** | 0.7178 | **0.6912** | 0.5833 | 0.7789 |
| Kim et al(Kim et al., 2019) | 0.8061 | 0.5175 | 0.6618 | 0.6234 | 0.6863 | 0.6549 | 0.4932 | 0.7434 |
| C2FNAS(Yu et al., 2020) | 0.8076 | 0.5441 | 0.6759 | 0.6430 | 0.7100 | 0.6765 | 0.5890 | 0.7697 |
| DiNTS(He et al., 2021) | 0.8102 | 0.5535 | 0.6819 | 0.6450 | 0.7176 | 0.6813 | **0.5912** | 0.7793 |
| AdwU-Net (ours) | **0.8172** | **0.5544** | **0.6858** | 0.6598 | **0.7226** | **0.6912** | 0.5621 | **0.7803** |

tasks including Heart, Liver, Hippocampus, Lung, Pancreas, and Hepatic Vessel. Overall, our methods achieve the best average Dice of 0.7803 in all methods without pre-training in the MSD leaderboard. The results show the power of designing task-specific architectural hyperparameters compared to using a fancy module. Some visualization comparisons are available in the Appendix.

Table 3: Quantitative results with average (standard deviation) of DSC on the MSD Prostate datasets by 5-fold cross-validation.

| Model | Peripheral Zone | Transition Zone | Average DSC |
|---|---|---|---|
| Baseline 3d nnU-Net | 0.6679 (0.2087) | 0.8410 (0.1477) | 0.7544 (0.1412) |
| **AdU-Net (depth search only)** | **0.6783 (0.2056)** | **0.8486 (0.1331)** | **0.7634 (0.1397)** |
| **AwU-Net (width search only)** | **0.6754 (0.2132)** | **0.8501 (0.1205)** | **0.7627 (0.1396)** |
| **AdwU-Net (depth and width search)** | **0.6882 (0.2018)** | **0.8595 (0.0810)** | **0.7738 (0.1179)** |

### 3.3. Depth Search versus Width Search

To evaluate the effectiveness of depth search and width search separately, we implement AdU-Net and AwU-Net which only search the optimal depth and width of the specific task respectively. For AdU-Net, the channel number in each resolution stage follows the setting of 3d nnU-Net and we only adopt the depth search. For AwU-Net, the convolution number in each stage is fixed to 2 which is the same as 3d nnU-Net and we only adopt the

width search. We compare their performance on the MSD Prostate dataset by 5-fold cross-validation and the result is shown in Table 3. Compared with the 3d nnU-Net baseline, depth search, width search and compound search improve the baseline by 0.90%, 0.83% and 1.94% in terms of the average dice of peripheral zone and transition zone, respectively. Paired T-test shows that the improvements are statistically significant at $p < 0.05$.

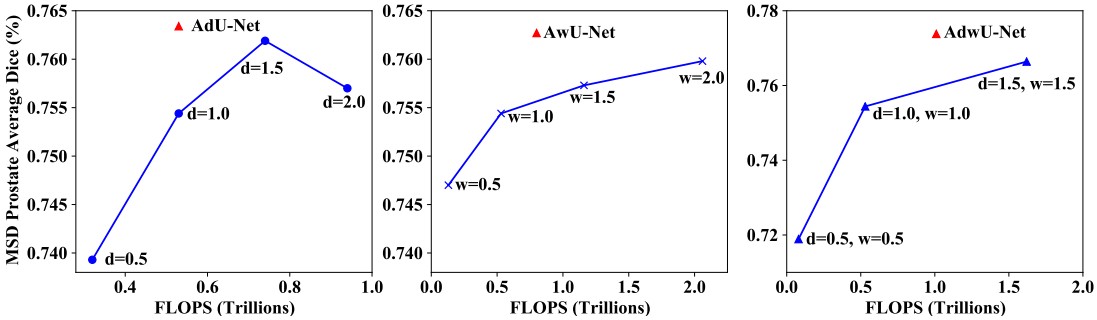

Figure 2: Scaling U-Net with different depth (d), width (w), and simultaneously. U-Net with our searcheded depth and width outperform scaled U-Net.

## 3.4. Compared to Model Scaling

To ensure that the performance gain is not simply due to the increment of parameters, we manually scale the depth and width of 3d nnU-Net and compare their performance with our searched models. Following (Tan and Le, 2019), we scale the baseline 3d nnU-Net with different depth coefficient $d$ of $[0.5, 1.0, 1.5, 2.0] \times$, width coefficient $w$ of $[0.5, 1.0, 1.5, 2.0] \times$, and simultaneously with coefficient of $[0.5, 1.0, 1.5] \times$. We report the results based on 5-fold cross-validation on the MSD Prostate dataset and use FLOPs as our metrics to compare resource consumption. As shown in Figure 2, with less computational costs, our searched models outperform the scaled models, which shows the effectiveness and efficiency of our proposed methods.

## 4. Conclusions

In this paper, we propose a new segmentation framework, which can efficiently complete the automatic search of task-specific optimal depth and width in vanilla U-Net. As far as we know, this is the first application of NAS to search the optimal depth and width simultaneously in the field of medical image segmentation. Experimental results show that on various datasets with different modalities, AdwU-Net can achieve consistently better performance than several state-of-the-arts searched by NAS methods and manually designed networks. In addition, our proposed Adwblock can easily be applied in other backbone networks for different tasks. As for future work, automatically designing the better configuration of other hyperparameters in the pipeline of medical image segmentation like input patch size can be considered.

## Acknowledgments

This research is partially supported by the National Key research and development program (No.2016YFC0106200), Beijing Natural Science Foundation-Haidian Original Innovation Collaborative Fund (No.L192006), and the funding from Institute of Medical Robotics of Shanghai Jiao Tong University as well as the 863 national research fund (No.2015AA043203). We thank the organization team of the MSD challenge (Antonelli et al., 2021) for the publicly available dataset. We also thank Fabian Isensee for his great PyTorch implementation of nnU-Net(Isensee et al., 2021).

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

## Appendix A. Comparison With More Methods

To compare with more methods without test results in the MSD leaderboard, we also compare our AdwU-Net with these methods in terms of 5-fold cross-validation results on the MSD Brain, Heart, and Prostate datasets. The results are presented in Table 4. These compared methods can be divided into two groups and all of them don't consider the optimal depth and width design of networks. The first groups contain U-Net and its variants designed manually. In the second group, these methods use the NAS method to design architectures automatically. Our AdwU-Net achieves better results than these methods while taking less search time than other NAS-based methods.

Table 4: Quantitative comparison between our method with manually designed architectures and auto-searched architectures on three subsets of the MSD dataset by 5-fold cross-validation. The first four rows are manually designed networks, the last row is our proposed method, others are NAS-based methods

.

| Model | Search Time | Brain Tumor | Heart | Prostate | Average |
|---|---|---|---|---|---|
| U-Net (Ronneberger et al., 2015) | - | 0.7250 | 0.9070 | 0.7313 | 0.7877 |
| ResU-Net (Zhang et al., 2018) | - | 0.7161 | 0.8960 | 0.6377 | 0.7500 |
| U-Net++ (Zhou et al., 2020) | - | 0.7266 | 0.9156 | 0.7295 | 0.7905 |
| 3d nnU-Net (Isensee et al., 2021) | - | 0.7411 | 0.9329 | 0.7544 | 0.8095 |
| RONASMIS (Bae et al., 2019) | 3.1 GPU Days | 0.7414 | 0.9272 | 0.7571 | 0.8085 |
| UXNet (Ji et al., 2020) | 3 GPU Days | 0.7457 | 0.9350 | 0.7636 | 0.8148 |
| **AdwU-Net (ours)** | **2 GPU Days** | **0.7467** | **0.9356** | **0.7738** | **0.8187** |

## Appendix B. Searched Architectures

We present the searched architectures on the MSD Brain, Heart and Prostate datasets in Figure 3, Figure 4, and Figure 5, respectively. We observe that the searched architecture on challenging datasets like Brain is deep and wide. In contrast, architecture searched on relatively easy datasets like Heart is shallow and narrow. This is consistent with our prior knowledge that we need more computational resources for difficult tasks as a deeper and

wider network can extract richer and higher-level features. For relatively easy tasks, using a shallower and narrower network is enough to handle, increasing the depth and width may lead to overfitting and cause performance degradation.

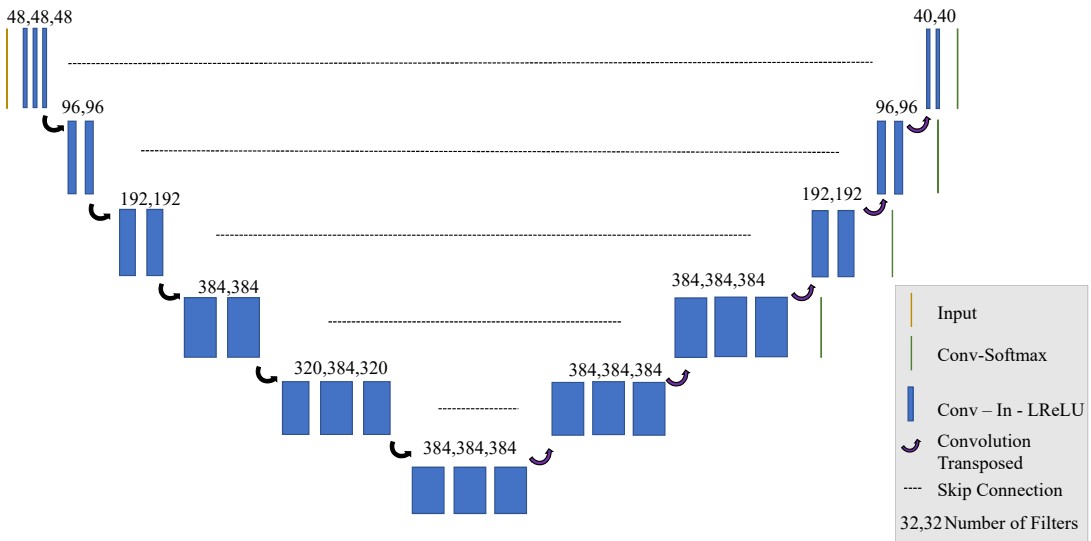

Figure 3: Searched architecture on the MSD Brain dataset

.

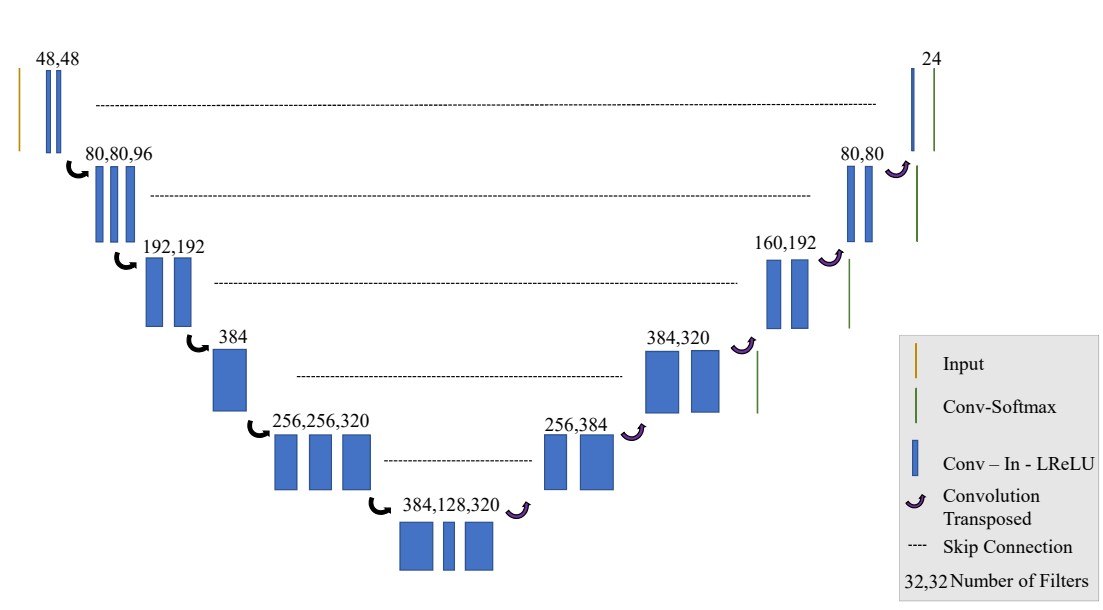

Figure 4: Searched architecture on the MSD Heart dataset

.

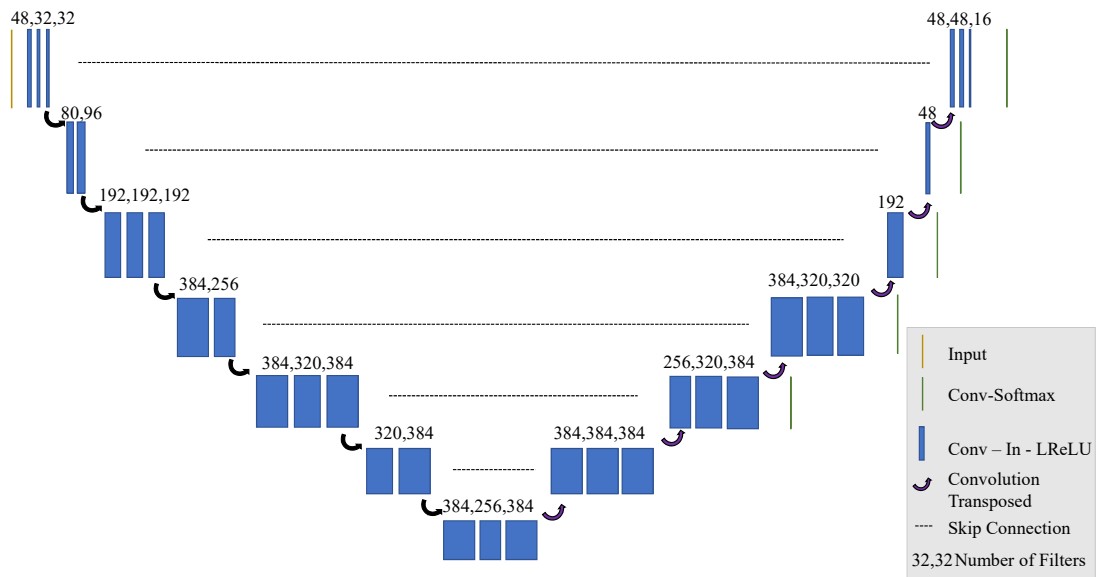

Figure 5: Searched architecture on the MSD Prostate dataset

.

## Appendix C. Compared to Model Scaling in Detail

Table 5 present the performance of scaled models and searched models in detail. We use FLOPs, model parameters, and GPU memory usage during training as our metrics to compare resource consumption of models.

Table 5: Quantitative comparison between our searched models and scaled models on the MSD Prostate datasets by 5-fold cross-validation.

| Model | FLOPs(T) | Parameters(M) | Memory(MB) | DSC |
|---|---|---|---|---|
| Scale 3d nnU-Net by depth ($d = 0.5$) | 0.32 | 26.23 | 10442 | 0.7393 |
| Scale 3d nnU-Net by depth ($d = 1.0$) | 0.53 | 42.50 | 11991 | 0.7544 |
| Scale 3d nnU-Net by depth ($d = 1.5$) | 0.74 | 60.01 | 12653 | 0.7619 |
| Scale 3d nnU-Net by depth ($d = 2.0$) | 0.94 | 81.27 | 15802 | 0.7599 |
| **AdU-Net (depth search only)** | **0.53** | **39.85** | **12178** | **0.7634** |
| Scale 3d nnU-Net by width ($w = 0.5$) | 0.13 | 8.19 | 4647 | 0.7470 |
| Scale 3d nnU-Net by width ($w = 1.0$) | 0.53 | 42.50 | 11991 | 0.7544 |
| Scale 3d nnU-Net by width ($w = 1.5$) | 1.16 | 70.28 | 15333 | 0.7573 |
| Scale 3d nnU-Net by width ($w = 2.0$) | 2.06 | 131.00 | 25160 | 0.7598 |
| **AwU-Net (width search only)** | **0.80** | **50.77** | **14324** | **0.7627** |
| Compound scale ($d = 0.5, w = 0.5$) | 0.08 | 4.85 | 3149 | 0.7189 |
| Compound scale ($d = 1.0, w = 1.0$) | 0.53 | 42.50 | 11991 | 0.7544 |
| Compound scale ($d = 1.5, w = 1.5$) | 1.62 | 98.96 | 19029 | 0.7664 |
| **AdwU-Net (depth and width search)** | **1.01** | **71.82** | **15717** | **0.7738** |

## Appendix D. Qualitative Comparison

We provide segmentation visualization from the MSD Brain Tumor, Heart and Prostate datasets. As shown in Figure 6, our AdwU-Net achieves overall better segmentation results.

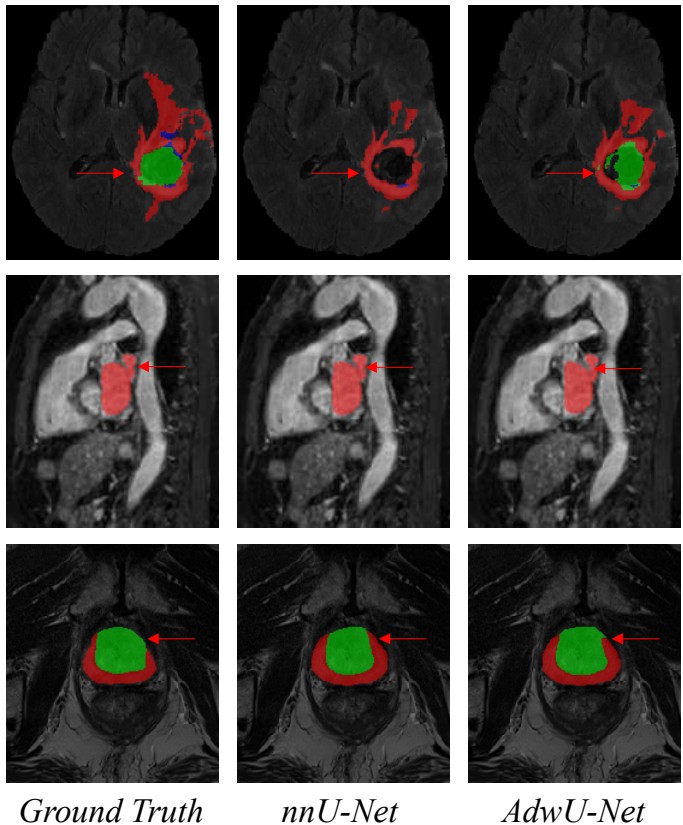

*Ground Truth*        *nnU-Net*        *AdwU-Net*

Figure 6: Qualitative comparison between the segmentation results of nnU-Net and our AdwU-Net on some challenging cases from three subsets of the MSD dataset

.

