# OpenReview forum: "AdwU-Net: Adaptive Depth and Width U-Net for Medical Image Segmentation by Differentiable Neural Architecture Search"
_MIDL.io/2022/Conference — MIDL 2022_

### Official Review · Reviewer_7xwu · 2022-01-23

**Confidence:** 2
**Preliminary Rating:** 4
**Recommendation:** Poster

**Summary:**

This paper proposed AdwU-Net: an efficient neural architecture search framework to search the optimal task-specific depth and width in the U-Net backbone. Experiments are conducted on three subsets of the MSD datasets where AdwU-Net shows better results than other methods with less number of GPU days required for optimization.

**Strengths:**

1) The problem explored in the paper is important as the proper design of U-Net architecture leads to a good performance.

2) The experiments conducted and the results reported seem to improve the performance while using lesser number of GPU days.

**Weaknesses:**

1) The authors mainly focus on searching for the best parameters to control the depth and width of UNet. There are two other interesting factors which also contribute the performance of UNet: Number of channels and feature resolution change (level of UNet). Why do the authors consider only the depth and width and not these characteristics? Are there any special reasons?

2) In terms of segmentation performance improvement, the improvement is not much when compared to nnUNet.

**Deanonymize Review:**

no

**Final Rating After The Rebuttal:**

4: Weak Accept

**Justification Of The Final Rating:**

The authors have responded saying that the number of channels is equal to the width of U-Net. I disagree to it. There are two things to consider with regard to width if you look at a standard UNet encoder block: 1) Number of conv layers at that level. 2) Number of filters in those con layers. Both these contribute to the width of the network as the conv layers operate on the same level. Otherwise, I find the rebuttal answer well to my queries.

**Paper Type:**

validation/application paper

**Questions To Address In The Rebuttal:**

I think this paper deserves to be accepted, so I do not think there is any point that I would want to be addressed for sure in the paper. However, answering the questions in weakness will help in better explain-ability of the paper.

**Special Issue:**

no

---

### Official Review · Reviewer_TXSy · 2022-01-24

**Confidence:** 3
**Preliminary Rating:** 2

**Summary:**

The authors propose the use of neural architecture search to find the optimal depth and width configurations of a U-Net model to perform image segmentation. They evaluate the performance of the searched architecture on a subset of the medical segmentation decathlon showing superior performance with the discovered parameters

**Strengths:**

The solution to finding the best possible network for medical image segmentation is a difficult problem to solve and the proposed solution appears promising. Additionally, the authors provide code along with the paper to make wider adoption easier.

**Weaknesses:**

Though results look promising, the organization and of the paper makes the key concepts difficult to understand. The results section lacks details on the baseline methods as well as the underlying tasks which are used to evaluate the model.

**Deanonymize Review:**

no

**Detailed Comments:**

Section 2.2.1 `y^s plays` - typo?
Section 2.2.2 is a crucial part of the paper, consider rewriting this in a clearer way
Section 2.3 Motivating the use of the two training datasets (A and B) would help here
 Algorithm 1 probably not required
Section 3.2  `State-of-the-Arts`, `Spells` - typos. More details on the SOTA methods chosen to be used would add value
Section 3.3 What is the key takeaway of this section? If it is just to show that depth and width search is better than depth or width search alone, I don’t think it adds much value.



**Final Rating After The Rebuttal:**

3: Borderline

**Justification Of The Final Rating:**

I would like to thank the authors for addressing the comments and concerns I had by updating the manuscript. I have updated my review to borderline as there are some aspects of the paper that I believe are of interest to the wider community.

**Paper Type:**

methodological development

**Questions To Address In The Rebuttal:**

Results are promising and the idea is interesting but the overall presentation of the paper lacks details in places. More work needs to be done on the results and methods section to better explain the key concepts of the paper and to help the reader better interpret the results.

**Special Issue:**

no

---

### Official Review · Reviewer_PwCS · 2022-01-24

**Confidence:** 3
**Preliminary Rating:** 4
**Recommendation:** Oral

**Summary:**

This paper applied a differentiable neural architecture search to develop a segmentation framework that efficiently searches the U-Net's optimal depth and width in a specific medical mission. The author compared the proposed method to the SOTA methods and conducted an ablation study to demonstrate the rationality of the architecture. The proposed framework, AdwU-Net, can efficiently achieve better performance than the manually designed networks and easily be used in other tasks.

**Strengths:**

1. This paper proposed an adaptive depth and width U-Net for medical image segmentation framework, which is memory-efficient and achieved notably performance.
2. The comparison experiments and ablation study shows that AdwU-Net achieves the SOTA performance in several medical image segmentation tasks.
3. The proposed method can use in other backbones for various tasks.

**Weaknesses:**

1. There is confusion in the symbols. In formula (1), $g_l^s$ denotes the Gumbel Softmax function of $l_{th}$ depth in stage $s$, and in formula (3), $g_i^s$ is the Gumbel weight parameter of the $i_{th}$ mask; In formula (2), the $o$ was not defined.
2. Some details are not precise. In section 2.3, the author did not explain how to use the $argmax$ operation. In section 3.2, the authors analyzed little about the results of the comparison methods. In Figure 1, there are 9 blocks in the figure, but there are 11 blocks used practically.
3. The experimental description and analysis are not detailed enough. The author can mention how to conduct comparison experiments. In addition, the author can note that visualization of the results has been written in the appendix or shown the visualization figure in the main text.

**Deanonymize Review:**

no

**Paper Type:**

both

**Questions To Address In The Rebuttal:**

1. Please improve the article's clarity, especially the formula and details. To be specific, check the formula (1), (2), and (3); explain the detail of how to use $argmax$ in section 2.3; try to revise Figure 1 if needed.
2. Please add some description of the comparison details and results.

**Special Issue:**

no

---

### Official Review · Reviewer_PgYm · 2022-01-24

**Confidence:** 4
**Preliminary Rating:** 4
**Recommendation:** Poster

**Summary:**

This paper proposes AdwU-Net, an efficient NAS framework to search the optimal task-specific depth and width in the U-Net backbone. Specifically, an adaptive depth and width block is designed and applied hierarchically in U-Net. In each block, the optimal number of convolutional layers and channels in each layer are directly learned from data. Also, this paper conducts an efficient architecture search and reuses the network weights of different depth and width options in a differentiable manner. Extensive experiments on three subsets of the MSD dataset show that the proposed method achieves better performances over other baselines.

**Strengths:**

1. The paper is well-written and adequately addresses prior work.

2. This paper proposes a new segmentation framework, AdwU-Net, which can efficiently complete optimal depth and width search for U-Net. And this paper designs a memory-efficient search space for depth and width search by feature map reusing, which enables exploring more depth and width options.

3. The proposed searched task-specific architectures are incorporated in the nnU-Net framework and achieves SOTA performance in several medical image segmentation datasets.


**Weaknesses:**

In Eq.4, this paper simply combines the cross entropy loss and dice loss with equal weights, while this is not always the best choice. The paper could introduce a balanced weight here and investigate the effect of the weight.

**Deanonymize Review:**

no

**Detailed Comments:**

The paper claims that the proposed method achieves better performances over SOTA baselines. While in Tab.2, only three NAS-based methods are compared, and the most recent one is UXNet (Ji et al., 2020). I would suggest the authors reorganize Tab.2. More specifically, try to remove a few of the manually designed networks (3d nnU-Net a necessary one while not for the others), and add more recent NAS-based methods as baselines.


**Final Rating After The Rebuttal:**

4: Weak Accept

**Justification Of The Final Rating:**

This paper proposes decent network architecture search based medical segmentation methods, and achieves SOTA performance in several medical image segmentation datasets. Aditionally, the authors somehow addressed my concerns, thus I'd like to keep my original rating.

**Paper Type:**

methodological development

**Questions To Address In The Rebuttal:**

1. Add an ablation study to inverstigate the effect of the balanced weight between the cross entropy loss and dice loss.

2. If possible, hopefully the authors could add more SOTA NAS-based methods as baselines to make this paper more solid. I don't want to name specific baselines here while any baselines from top venues of CV/Medical Imaging in recent years will be okay.

**Special Issue:**

no

---

### Meta-Review · Area_Chair_aanD · 2022-02-20

**Recommendation:** Accept (Poster)
**Confidence:** 5

**Metareview:**

After rebuttal, the manuscript receives 3 weakly accept and 1 borderline. The authors have clearly clarified the issues raised by the reviewers. Overall, Most reviewers are satisfied with the response given by the authors and are glad to see that the quality of the paper has been improved substantially.

---

### Decision · Program_Chairs · 2022-02-28

Accept